# Osteoclastogenic Potential of Tissue-Engineered Periosteal Sheet: Effects of Culture Media on the Ability to Recruit Osteoclast Precursors

**DOI:** 10.3390/ijms22042169

**Published:** 2021-02-22

**Authors:** Kohya Uematsu, Takashi Ushiki, Hajime Ishiguro, Riuko Ohashi, Suguru Tamura, Mari Watanabe, Yoko Fujimoto, Masaki Nagata, Yoichi Ajioka, Tomoyuki Kawase

**Affiliations:** 1Division of Dental Implantology, Niigata University Medical and Dental Hospital, Niigata 951-8520, Japan; ue@dent.niigata-u.ac.jp; 2Department of Transfusion Medicine, Cell Therapy and Regenerative Medicine, Niigata University Medical and Dental Hospital, Niigata 951-8520, Japan; tushiki@med.niigata-u.ac.jp (T.U.); mwatanabe@med.niigata-u.ac.jp (M.W.); yfujimoto@med.niigata-u.ac.jp (Y.F.); 3Department of Hematology, Endocrinology and Metabolism, Faculty of Medicine, Niigata University, Niigata 951-8510, Japan; power@med.niigata-u.ac.jp (H.I.); s.tamura911@gmail.com (S.T.); 4Histopathology Core Facility, Faculty of Medicine, Niigata University, Niigata 951-8510, Japan; riuko@med.niigata-u.ac.jp (R.O.); ajioka@med.niigata-u.ac.jp (Y.A.); 5Division of Molecular and Diagnostic Pathology, Graduate School of Medical and Dental Sciences, Niigata University, Niigata 951-8510, Japan; 6Division of Oral and Maxillofacial Surgery, Graduate School of Medical and Dental Sciences, Niigata University, Niigata 951-8514, Japan; nagatam@dent.niigata-u.ac.jp; 7Division of Oral Bioengineering, Graduate School of Medical and Dental Sciences, Niigata University, Niigata 951-8514, Japan

**Keywords:** periosteal sheets, green fluorescent protein, bone marrow transplantation, CD11b, osteoclastogenesis, tartrate-resistant acid phosphatase

## Abstract

Cell culture media influence the characteristics of human osteogenic periosteal sheets. We have previously found that a stem cell medium facilitates growth and collagen matrix formation in vitro and osteogenesis in vivo. However, it has not yet been demonstrated which culture medium is superior for osteoclastogenesis, a prerequisite for reconstruction of normal bone metabolic basis. To address this question, we compared chemotaxis and osteoclastogenesis in tissue-engineered periosteal sheets (TPSs) prepared with two types of culture media. Periosteal tissues obtained from adult volunteers were expanded with the conventional Medium 199 or with the stem cell medium, MesenPRO. Hematopoietic enhanced-green-fluorescent-protein (EGFP)-nude mice were prepared by γ-irradiation of Balb/c nu/nu mice and subsequent transplantation of bone marrow cells from CAG-EGFP C57BL/6 mice. TPSs were implanted subcutaneously into the chimeric mice and retrieved after intervals for immunohistopathological examination. EGFP^+^ cells were similarly recruited to the implantation site in both the TPSs prepared, whereas the distribution of CD11b^+^ cells was significantly lower in the TPS prepared with the stem cell medium. Instead, osteoclastogenesis was higher in the TPS prepared with the stem cell medium than in the one prepared with the conventional medium. These findings suggest that the stem cell medium is preferable for the preparation of more functional TPSs.

## 1. Introduction

Human tissue-engineered periosteal sheets (TPSs) are osteogenic biomaterials that can be used for bone regeneration. They were originally proposed by O’Driscoll in the 1990s [1] and were subsequently modified by the addition of the in vitro expansion process, and were tested in an animal study by Mizuno et al. using a canine model [2]. After this proof-of-concept, we made some further modifications to optimize this protocol for human alveolar periosteum tissue and performed the first-in-human trial of periodontal regeneration [3,4,5]. To date, regenerative therapies using TPSs have been applied in more than 120 cases for both periodontal regeneration and alveolar ridge augmentation. The clinical outcome of the TPS is more predictable and potent than that of other conventional bone substitutes, and its remarkable advantage is that no adverse effects have been identified upon its use.

However, the bone formation mechanism is poorly understood. Our previous in vitro and in vivo studies [6,7,8,9,10] clarified that TPSs can induce osteoblastic differentiation of the cells in the TPSs, mineral deposit formation in vitro, and form bonelike tissue in vivo. Simultaneously, an animal study showed enhanced angiogenesis and osteoclast formation at the ectopic site [6]. Taken together, these results suggest that both osteogenesis and angiogenesis and subsequent osteoclastogenesis may be cooperatively involved in TPS bone regeneration. In contrast to osteogenesis, osteoclastogenesis has rarely been investigated. According to the literature [11,12], osteoclast precursor cells, such as monocytes, are recruited to bone tissue and fuse to form osteoclasts with the aid of osteoblasts. Thus, similar scenarios could be applied to bone regeneration with TPSs. However, there has been no direct evidence as to which cell type(s) are recruited for osteoclastogenesis and which cell source(s) provide those cells, surrounding tissue, or circulating systems. 

In this study, we developed a novel approach to address these questions. In brief, unique nude mice with bone marrow (BM) cells expressing enhanced green fluorescent protein (EGFP) were developed by bone marrow transplantation (BMT). Different types of TPSs were produced using two different types of culture media: the first media used was the classic conventional medium, Medium 199 supplemented with 10% fetal bovine serum (FBS), which induces mild spontaneous osteoblastic differentiation [7,8,9]. The second was a stem-cell culture medium MesenPRO-RS™ (Invitrogen) supplemented with 2% FBS, which was originally designed and developed to expand stem cells without losing their stemness through spontaneous differentiation. This ability was confirmed in the explant culture of the TPS. This stem cell medium potently induces the growth of TPSs and simultaneously suppresses the spontaneous differentiation observed in conventional media [7]. Our hypothesis regarding the possible mechanisms behind TPS-induced osteoclastogenesis is illustrated in Figure 1, where recruitment of osteoclast precursor cells and osteoclastogenesis is believed to be mainly influenced by angiogenic, chemotactic, and osteoclastogenic factors. In parallel, we also aimed to examine the validity and usefulness of the experimental design using an immunodeficient animal model to study the osteoclastogenic ability of implanted TPSs.

## 2. Results

### 2.1. In Vitro Studies

Small pieces of periosteum tissues were cultured using two types of culture media, Medium 199 supplemented with 10% FBS and MesenPRO supplemented with 2% FBS, which resulted in TPSs at different differentiation stages of osteoblastic cells. As described in a previous study [7], we reconfirmed that Medium 199, which is used as a standard culture medium for TPS preparation for clinical use [3,5,13,14], upregulates alkaline phosphatase (ALP) activity and biomineralization ability (Figure 2). MesenPRO medium, which was originally developed for expansion of human mesenchymal stem cells (MSCs) while maintaining a multipotential phenotype in TPSs [7], showed significantly greater potency on the growth of TPSs, but suppressed their differentiation along the osteogenic cell lineage (Figure 2). We also reconfirmed that compared to Medium 199, MesenPRO medium potently built cell-multilayers up to form thicker cell sheets, with enriched extracellular matrices (mainly collagen), as demonstrated previously [7].

To analyze the ability to recruit hematopoietic cells to the implanted TPSs, chemokine expression levels were determined using qPCR in TPSs after an eight-week culture (Figure 3). Stromal cell-derived factor-1 (SDF-1), chemokine C-C motif ligand (CCL2), and chemokine C-C motif ligand (CCL5) expression levels were significantly higher in the TPSs prepared with Medium 199 than in those prepared with MesenPRO medium. The differences in chemokine C-C motif ligand (CCL1), chemokine C-C motif ligand (CCL3), and chemokine C-C motif ligand (CCL4) expression levels were not statistically significant between the two groups, but were somewhat higher in the conventional medium group than in the stem cell medium group. Thus, in vitro chemokine expression levels were generally high in the TPSs prepared with the conventional medium.

### 2.2. In Vivo Studies

To examine the possible assisting effects of GFP^+^ hematopoietic cells on calcification in TPSs, BM mononuclear cells (BM-MNCs) from CAG-EGFP C57BL/6 mice were transplanted into Balb/c nu/nu recipients. In these mice, only hematopoietic cells, including BM, spleen, lymph nodes, thymus, and peripheral blood express EGFP proteins. Thus, we established hematopoietic cell-specific EGFP-expressing Balb/c nu/nu mice (hematopoietic EGFP-nude mice). In this model, hematopoietic cell-derived cells, including osteoclasts, are anticipated to be EGFP-positive if their origin is from the engrafted donor cells. Experimental mice were highly reconstituted (>90% H-2^b+^ in the blood) and both types of TPSs were subcutaneously transplanted to the backs of the mice. In these mice, hematopoietic cells were well reconstituted in the peripheral blood five weeks after BMT (Appendix A). The transplanted TPSs were excised together with the surrounding tissues and analyzed at eight weeks post transplantation.

Ectopic bonelike tissue was formed in both the TPSs, prepared with the conventional and the stem cell media, with medium to high probabilities of 46.2% and 75.0%, respectively (Figure 4a, Table 1). Fisher’s exact test revealed no statistical difference between these two TPS types. As a negative control, a cell-free collagen sponge was adopted, but no calcified tissue formation was observed in it (data not shown).

Compared to bonelike tissue formation, the probability of tartrate-resistant acid phosphatase (TRAP)-positive cell induction was lower in both the media (Figure 4b, Table 2). However, the percentages of single- and multinucleated TRAP^+^ cell induction were 15.4% and 50.0% in the conventional and the stem cell media, respectively. Fisher’s exact test revealed that the stem cell medium was more potent in this induction than the conventional medium.

As expected, infiltration of EGFP^+^ cells was observed around both types of TPSs, but no significant difference was observed between these TPSs (Figure 5a,c). To determine the phenotype of EGFP^+^ cells in the TPSs, the infiltration of CD11b^+^ cells was further examined. CD11b^+^ cells were distributed similarly in and around TPSs as GFP^+^ cells; however, the infiltration of CD11b^+^ cells was significantly higher in the TPS prepared with the conventional medium than in the one prepared with the stem cell medium (Figure 5b,c). The CD11b^+^ cells pathologically consisted of monocytes and not granulocytes.

## 3. Discussion

### 3.1. General Outlook

In general, cells that are used for tissue regenerative therapy should be well differentiated with high purity [15]. Although it is challenging to purify and homogenize all periosteal cells contained in TPSs to maturated osteoblastic cells, according to the widely accepted concept mentioned above, we believe that TPSs expressing higher ALP activity and mineralizing activity will serve as better grafting materials for bone regeneration. However, taken together with the previous data [7,16], the present data correct that concept and, in addition, suggest that a highly cell-multilayered, but less mature, TPS could serve as a better bone grafting material.

It is well known that the growth and maintenance of bone volume is controlled by the balance between bone formation and resorption [17]. Despite such a biologically exquisite homeostatic control, most attention in bone regenerative therapy has been predominantly paid to X-ray-sensitivity and bio-/nonbiomineralization. This partiality may be due to technical limitations with respect to bone regeneration and the primary medical requirement for mechanical stress. Thus, such a X-ray-based positive evaluation of bone regeneration does not necessarily indicate that the ideal homeostatic balance is restored. For example, although considered as the most successful bone substitute [18], β-tricalcium phosphate has a few disadvantages. One is its weak mechanical properties, and the other is its faster absorption [19]. These disadvantages imply that the resorption, especially initial resorption, is not mediated by osteoclasts and that mineralization is not fully dependent on osteoblasts. It can be thought that chemical dissolution and deposition are involved in this process without control by homeostatic balance. Thus, we believe that ideal bone regeneration is to restore the balance between biomineralization and bioresorption and that this is the ultimate goal of bone regenerative therapy.

In our clinical studies and trials, in which TPSs were applied along with platelet-rich plasma (PRP) and reduced amounts of autologous crushed bone [13,14], the bone homeostatic balance was regained much faster in the TPS group than in the group using autologous crushed bone alone. In the animal study, both PRP and crushed bone were excluded to focus on the potential of TPS, by avoiding the influence of factors derived from these materials. However, considering the data obtained from our basic research [6,7,8], we hypothesized that the TPS alone is effective enough to induce osteoclastogenesis; this ability of the TPS could also serve as a criterion for its quality. Thus, from the viewpoint of tissue engineering, this ability could be useful for optimization and standardization of processing protocols.

In this study, we compared two types of TPSs: one was processed with conventional Medium 199 supplemented with 10% FBS, while the other was processed with stem cell medium, MesenPRO supplemented with 2% FBS. The latter medium is nutritionally good enough to expand stem cells without FBS; however, FBS, especially the adhesion factors served by FBS, are necessary for TPS processing, to maintain pieces of periosteum tissue statically at the same position and facilitate cell migration from inside to outside the pieces. The resulting TPSs processed with the conventional and stem cell media were characterized as relatively mature and immature TPSs, respectively, based on the levels of ALP activity and biomineralization in vitro.

### 3.2. Recruitment of BM Cells

In this study, the mRNA expression levels of chemotactic factors indicated that the TPS prepared with the conventional medium would display higher ability to recruit EGFP^+^ BM cells than the one prepared with the stem cell medium. As expected, the ability of neovascularization, which is expected to support BM cell recruitment in the nude mouse, was higher in the TPS prepared with the conventional medium than in the one prepared with the stem cell medium. However, at eight weeks post implantation, the infiltration of EGFP^+^ cells were not statistically different between these two TPS types. In addition, although the inflammation may be in transition from the acute to the chronic phase at this time-point, these cells were morphologically identified as cells in the monocyte lineage (monocytes and macrophages) in both the cases.

The possibility that mesenchymal stem cells (MSCs) are also EGFP^+^ and recruited as precursors of osteogenic or endothelial cells to the implantation site cannot be ruled out. However, as far as we examined, we could not find EGFP^+^ cells that could morphologically be identified as osteoblasts or vascular endothelial cells, in or around the implanted TPSs. Taken together, the present data suggest that most EGFP^+^ cells are monocytic cells that can be utilized for osteoclast formation under these conditions and that this ability is significantly higher in the TPS prepared with the stem cell medium.

### 3.3. Osteoclast Formation

Matured osteoblasts are known to facilitate osteoclastogenesis through secretion of colony-stimulating factor-1 (CSF-1), also known as macrophage-colony stimulating factor (M-CSF), and receptor activator of nuclear factor-kappa B ligand (RANKL) [20]. In a previous study [6], we confirmed that post subcutaneous implantation, the TPS receiving pretreatment for osteoblast induction showed higher angiogenic and osteoclastogenic abilities than the nonpretreated TPS. However, in this study, we observed that despite the immature stages of osteogenesis, the TPS prepared with the stem cell medium acquired higher osteogenic potential post implantation. Thus, the abovementioned phenomenon could be explained by the possibility that sufficient quality and quantity of immature osteoblast progenitors and precursors are prepared by in vitro expansion and differentiate into mature osteoblasts immediately after implantation. The extracellular matrices (ECMs) and collagen, which had been enriched in the TPS prepared with the stem cell medium [7], may also contribute to this immediate differentiation by providing osteogenesis-inducing factors, such as transforming growth factor-β (TGF-β), involved in osteoblast proliferation and bone formation [21,22]. Although bone morphogenetic proteins (BMPs), members of the TGF-β family that potently induce osteogenesis, do not form latent complexes in ECMs, BMP-2 is known to be bound to ECMs in vivo [23]; this has often been applied during the development of BMP delivery systems [24]. Our preliminary DNA microarray examination, performed in parallel with a previous study [25], demonstrated that the TPS prepared with the stem cell medium upregulated the mRNA level of M-CSF (approximately 3-fold) and several isotypes of bone morphogenetic protein (BMP) (BMP-6, -7, and -8) (approximately 2.9–6.6-fold) two weeks postimplantation, when compared with the TPS prepared with the conventional medium (Kawase, unpublished observations).

CD11b, an α chain of the leukocyte β2-integrin, is generally thought to be a requisite for osteoclastogenesis [26,27]. This surface antigen is a representative marker of cells of the myeloid series, such as those with monocyte and granulocyte lineage, and is upregulated with progress in differentiation [28] but downregulated after fusion to form osteoclasts [29,30]. To evaluate the ability of osteoclastogenesis, in the present study, we determined not only osteoclast counts but also CD11b^+^ cell counts, in and around the implanted TPS, to provide supporting data for osteoclastogenesis. Osteoclastogenic ability was higher in the TPS prepared with the stem cell medium, than in the one prepared with the conventional medium. In support of this finding, CD11b^+^ cell counts were lower in the TPS prepared with the stem cell medium than in the one prepared with the conventional medium. On the other hand, there have been several recent research and review articles which believe that the established theory should be modified. One demonstrated that CD11b functions as a negative regulator of the earliest stages of osteoclast differentiation [30], while the other one described more diverse mechanisms behind osteoclastogenesis; under pathological conditions, differentiated monocytes and dendritic cells differentiate into osteoclasts, while in the steady state, BM progenitors form osteoclasts [31].

In this study, many myeloid progenitors that can be recruited to TPSs exist in the BM. In addition, the surgical operation for TPS implantation causes mild inflammation, even though the animal model used was an immunodeficient mouse. In addition, there were no signs of skin infection; thus, the CD11b^+^ cell infiltration is probably not due to infection. Taken together, CD11b^+^ monocytic cells are recruited by TPSs and serve as a major cell source of osteoclasts that the TPS acts on during ossification.

### 3.4. Characterization and Limitations of Bone Marrow (BM) Chimeric Nude Mice

The nude mouse is an immunodeficient animal that can accept human-derived tissues without severe immune rejection. For this reason, we have adopted this animal model to study the osteogenic ability of TPSs in a series of our studies [6,7,8,9,16,32,33]. While the nude mouse has B lymphocytes, monocytes, neutrophils, and other major blood cells derived from hematopoietic stem cells, it lacks post-thymic mature T lymphocytes, due to a lack of the thymus [34]. Therefore, the implanted TPS cannot be severely attacked by the immunological systems remaining in the animal model, because of the lack of mature T cells. At this point, this animal model may not be best suitable for studying the involvement of BM cells in the osteogenecity of TPS; however, because rodent-derived periosteum tissue pieces cannot form cell-multilayered sheets in vitro (Kawase, unpublished observation) and because the second purpose of this study was to develop a unique method to evaluate the ability of human-derived TPSs to form bones with sound metabolic balance, we examined the validity and usefulness of this experimental design.

The blood cell types infiltrating the implanted TPS were morphologically identified as monocytes, but not lymphocytes. Although we observed for eight weeks post implantation, a period that is generally thought to be sufficient for transition from initial acute inflammation to chronic inflammation, we did not observe lymphocytic infiltration. Charles River reported that lymphocytes are majorly white blood cells (approximately 63%), as compared to monocytes (approximately 3.6–4.1%), in the peripheral blood of Balb/c nude mice aged 8–10 weeks [35]. In an additional experiment, we also obtained similar percentages of blood cell counts. Thus, considering the blood cell counts, the ability of TPS to recruit monocytes is more dominant than that of lymphocytes. Interestingly, the recruitment of monocytes is also more dominant than that of granulocytes, probably due to upregulation of CCLs. In our experiment, chemokine levels were easily changed by the culture condition, especially in CCL1, CCL2 and CCL5. In fact, CCL1,2,5 is reported to cause strong association for recruitment monocytes/macrophages in immunological diseases such as rheumatoid arthritis and cardiovascular diseases [36,37].

Regarding the origin of the murine hematopoietic cells, we successfully achieved hematopoietic reconstitution by the transplanted BM-derived hemopoietic stem cells, and donor chimerism reached greater than 90% in peripheral blood. As for MSCs, there are no convincing data available in the literature, and we could not observe the possible recruitment of EGFP^+^ cells in the osteoblast lineage. In a parallel study using the same experimental design, we performed DNA microarray examination but did not find that osteoblasts are differentiated from mouse-derived cells. Thus, it seems less plausible that either type of TPS is capable of significantly recruiting or inducing osteoblasts from BM or from surrounding tissues, respectively.

In this study, we obtained donor BM cells, the majority of which were hematopoietic cells, using density gradient centrifugation with Ficoll^®^ fractionation. Therefore, it is speculated that donor MSCs are rarely included in the BM mononuclear cells prepared for transplantation. However, the possible inclusion of MSCs cannot be excluded. Because skeletal growth was observed for up to 16 weeks post BM cell transplantation, although their origin cannot be identified between the donor and the recipient, it can be thought that BM-derived MSCs are alive and involved in bone formation. Because we could not morphologically identify EGFP^+^ osteoblastic cells around the bonelike tissue in the implanted TPS, we find the possibility that the mesenchymal stem/progenitor cells are recruited from the BM by the TPS, to contribute to osteogenesis, as rare. This is a major limitation of this experimental design.

An additional limitation is the possible suppression or sensitization of the grafted BM cells. In BM cell transplantation, the grafted stem cells migrate from the peripheral blood and are successfully anchored in the BM during the homing process, after which they start proliferation in the subsequent engraftment process [38,39]. Therefore, it is necessary to wait for a time-period before performing the TPS implantation experiment, to exclude the possibility of engraftment failure. In mice, the hematopoietic reconstitution process usually takes at least 3–4 weeks. In this study, we took five weeks to observe the stabilization of the engrafted cells as previously described from our group [40]. By doing so, we confirmed that this limitation was excluded and minimized in this study.

As described above, this animal model has some non-negligible limitations. However, considering that the hematopoietic stem cells were replaced in high percentages (>90%) in this study, this animal model has a remarkable advantage in effectively demonstrating the recruitment of cells differentiated along the monocyte lineage and facilitating osteoclast formation at the implantation site. Therefore, we believe that due to its distinguishable advantages, this model can serve as a validated model for specific purposes, such as in the investigation of the involvement of infiltrated or residential CD11b^+^ cells in osteoclastogenesis.

## 4. Clinical Relevance

The TPS is a tissue-engineered bone grafting material that can be prepared with minimum manipulation. Thus, the major rate-limiting step in its preparation is the processing period. When TPS is prepared with the conventional medium, the processing period is six weeks long; the use of the stem cell medium can shorten this period. Despite suppressing in vitro osteoblast differentiation, the stem cell medium could still potentialize the TPS into an osteogenic bone grafting material. Taken together with our previous findings regarding the usefulness of the extract of platelet-rich fibrin (PRFext) [41], the stem cell medium is expected to further increase the safety of the TPS, in cooperation with the autologous serum substitute PRFext. Compared to other conventional bone substitutes, regenerated bone is expected to have a better homeostatic balance between bone formation and resorption. This will be greatly beneficial to all patients, clinicians, and hospitals possessing cell-processing facilities.

## 5. Materials and Methods

### 5.1. Preparation of Tissue-Engineered Periosteal Sheets (TPSs)

After obtaining informed consent, periosteum tissue segments were aseptically dissected from the periodontal tissue of the buccal side of the retromolar region in the mandible of 13 volunteers (4 males and 9 females; age: 19–35). Explant culture was performed as described previously [7,16,41]. In brief, small pieces of periosteum were placed on 60-mm dishes and subjected to explant culture in a 5% CO_2_-humidified and 95% air at 37 °C using the following two culture methods: Medium 199 (Invitrogen, Carlsbad, CA, USA) supplemented with 10% FBS (Invitrogen) or MesenPRO-RS™ (Invitrogen) supplemented with 2% FBS, in the presence of 25 μg/mL ascorbic acid 2-phosphate and antibiotics, for 8 weeks. The culture medium was refreshed twice weekly.

All the subjects enrolled in this study provided informed consent. The study protocol was approved by the Ethics Committee for Human Subject Use at Niigata University (# 2015–2288). All the animal experiments in this study were also performed with the approval of the Animal Ethics Committees of Niigata University (SA00176 and SA00654) and that of the Safety Committee for Recombinant DNA Experiments of Niigata University (SD00391 and SD01275).

### 5.2. Bone Marrow Transplantation (BMT)

CAG-EGFP transgenic mice on a C57BL/6 background (H-2^b^) and Balb/c nu/nu mice (H-2^d^) were obtained from Japan SLC Inc. (Hamamatsu, Japan), and housed for 1 week prior to the experiment. Care and use of the animals were in accordance with the Guiding Principles for the Care and Use of Animals approved by Niigata University Animal Ethics Committee. BMT was performed as previously described [40]. In brief, BM cells were collected from the medullary cavities of the humerus, femur, and tibia. BM-MNCs were obtained using density gradient centrifugation with Lympholyte-M solution (Cedarlane Labs, Hornby, ON, Canada). For BMT, recipient Balb/c nu/nu mice (8–10 weeks old) were sublethally irradiated (6.5 Gy) in a single fraction with a ^137^Cs PS-3000SB irradiator (Pony Industry Co. Ltd., Osaka, Japan). Following that, 3 × 10^6^ donor BM-MNCs from CAG-EGFP mice (8–10 weeks old) were injected into the recipient mice through the tail vein, 3–4 h post irradiation. Hematopoietic chimerism was measured 5 weeks post BMT.

### 5.3. Hematology and Flow-Cytometry

Cell counts were performed on at 5 weeks post BMT. Flow cytometric analysis was performed using CytoFLEX (Beckman Coulter, Brea, CA, USA). Hematopoietic chimerism was also measured 5 weeks post BMT using anti-H-2^b^ (AF6-88.5) and H-2^d^ (SF1-1.1) antibodies sourced by BioLegend (San Diego, CA, USA) [42].

### 5.4. Tissue-Engineered Periosteal Sheet (TPS) Implantation into the Bone Marrow-Transplanted Mice

Implantation of human TPSs into BMT mice was performed at 6 weeks post BMT. TPSs were harvested from culture dishes using a cell scraper and implanted into the dorsal subcutaneous tissue of BMT mice. Surgical procedures were performed aseptically, but without the aid of antibiotics, as previously described [7,16]. Eight weeks post TPS implantation, TPSs were harvested from the dorsal back of BMT mice, with the surrounding tissue.

### 5.5. Quantitative Real-Time Polymerase Chain Reaction (qPCR)

Total RNA was extracted from human TPSs collected from culture plates after 8 weeks of culture, using RNeasy^®^ Mini Purification Kit (Qiagen, Germantown, MD, USA), according to the manufacturer’s instructions. Reverse transcription was performed using the SuperScript VILO™ complementary DNA Synthesis Kit (Invitrogen, Carlsbad, CA, USA). Quantitative PCR was performed using a StepOne™ Real-Time PCR System (Applied Biosystems, Carlsbad, CA, USA) and TaqMan^®^ probes were used for TaqMan^®^ Gene Expression Assays (Applied Biosystems, Foster City, CA, USA): human CCL1 (Hs00171072-m1), CCL2 (Hs00234140-m1), CCL3 (Hs00234142-m1), CCL4 (Hs99999148-m1), CCL5 (Hs00982282-m1), SDF-1 (Hs03676656_mH), and human glyceraldehyde-3-phosphate dehydrogenase (Hs99999905_m1).

### 5.6. Histological and Immunohistochemical Examination

TPSs implanted in BMT mice were retrieved with their surrounding tissue, fixed in 10% neutral-buffered formalin, decalcified with 0.5 M ethylenediaminetetraacetic acid and embedded in paraffin. The blocks were sectioned to a thickness of 3 µm for hematoxylin and eosin staining. For immunostaining of EGFP and CD11b, the sections were deparaffinized and heat-induced epitope retrieval was performed at 121 °C for 20 min in pH 6.0 buffer (Nichirei biosciences, Tokyo, Japan) for the detection of CD11b. Endogenous peroxidase of the sections was blocked using 3% hydrogen peroxidase-methanol for 15 min, followed by incubation of the sections with 10% normal goat serum in PBS. For detecting EGFP^+^ cells, the sections were incubated with chicken anti-EGFP antibody (ab13970; at 1:500 dilution, Abcam, Cambridge, UK) at 4 °C overnight and then with horseradish peroxidase (HRP)-conjugated goat anti-chicken IgY (Ab97135, at 1:500 dilution, Abcam) for 1 h at room temperature. For CD11b staining, the sections were incubated with rabbit anti-CD11b (ab133357; at 1:100 dilution, Abcam) at 4 °C overnight, followed by incubation with a Histofine^®^ Simple Stain mouse MAX PO^®^ Kit (Nichirei biosciences) for 1 h at room temperature. Immunoreactions were visualized using a Histofine^®^ diaminobenzidine (DAB) Substrate Kit (Nichirei bioscience), followed by counterstaining of the sections with hematoxylin.

To detect osteoclasts, TRAP staining was carried out according to the method described by Burstone [43] with some modifications. Briefly, a mixture of 20 mg of naphthol AS-BI phosphate (Sigma-Aldrich, St. Louis, MO, USA), 30 mg of Fast Red ITR Salt (Sigma-Aldrich), and 50 mM sodium tartrate (Wako Pure Chemicals, Osaka, Japan) diluted in 0.1 M sodium acetate buffer (pH 4.8) was added to the deparaffinized sections and incubated for 3 h at room temperature. The sections were counter-stained with hematoxylin.

In addition, for assessment of ALP activity and Alizarin Red S staining, cultured TPSs were fixed in 10% neutralized formalin on dishes and stained using an ALP-staining kit (Muto Chemicals, Tokyo, Japan) or with 40 mM Alizarin Red S (Wako Pure Chemicals) for calcium deposits [8].

### 5.7. Image Analysis

For EGFP^+^ and CD11b^+^ cells, the corresponding positive areas were quantified and normalized by the total area using WinROOF software (version 6.0; Mitani Corp., Fukui, Japan), as described previously [44,45].

### 5.8. Statistical Analysis

Most data are expressed as mean ± standard deviation. A nonparametric Mann–Whitney U test was performed to test statistical differences, using the statistical software BellCurve for Excel (Social Survey Research Information Co. Ltd., Tokyo, Japan). Data for the presence (i.e., positive) of new bone formation and TRAP^+^ cells in individual specimens were statistically analyzed using Fisher’s exact test. A *P* value of less than 0.05 was considered to indicate statistically significant difference.

## 6. Conclusions

The TPS prepared with the conventional medium has already provided predictable and convincing clinical outcomes in alveolar ridge augmentation and periodontal regenerative therapy [4,5,13,14]. Our previous findings obtained from basic research suggest that modification of the culture medium, by including serum-substitute supplements, can shorten the processing periods [7,41]. The findings of the present study suggest that switching the culture medium to stem cell medium could further potentiate the TPS to regain the homeostatic metabolic balance between bone formation and resorption at the implantation site within relatively shorter periods of time, which might result in a higher-quality bone.

## Figures and Tables

**Figure 1 ijms-22-02169-f001:**
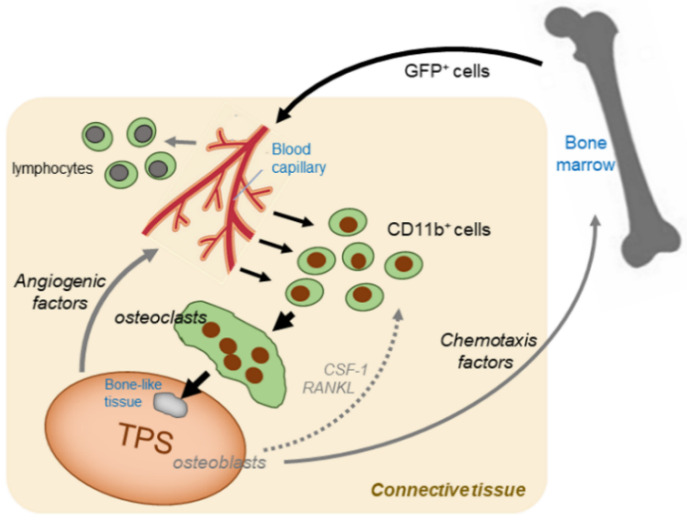
An illustration depicting possible mechanisms of tissue-engineered periosteal sheet (TPS)-induced osteoclastogenesis.

**Figure 2 ijms-22-02169-f002:**
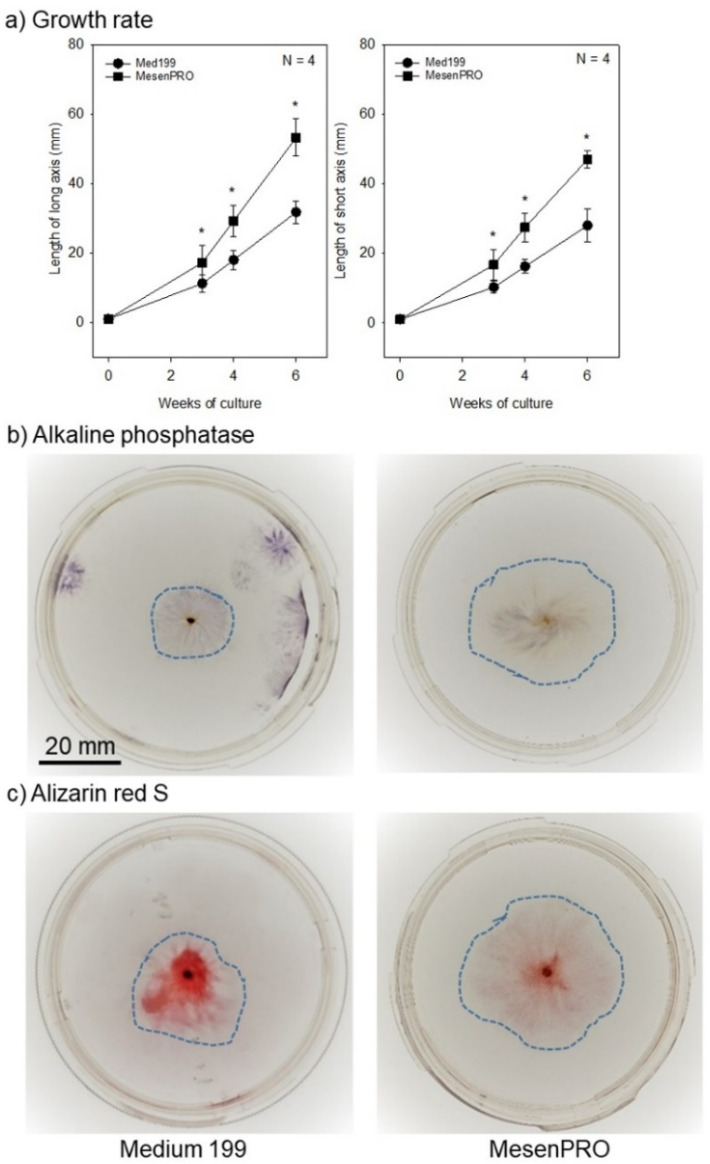
Effects of culture media on the growth and osteoblast differentiation of tissue-engineered periosteal sheets (TPSs) in vitro. (**a**) Growth rates, (**b**) alkaline phosphatase activity staining, and (**c**) Alizarin Red S staining for biomineralization. The broken lines represent the edges of TPSs. * *p* < 0.05 vs. Medium 199.

**Figure 3 ijms-22-02169-f003:**
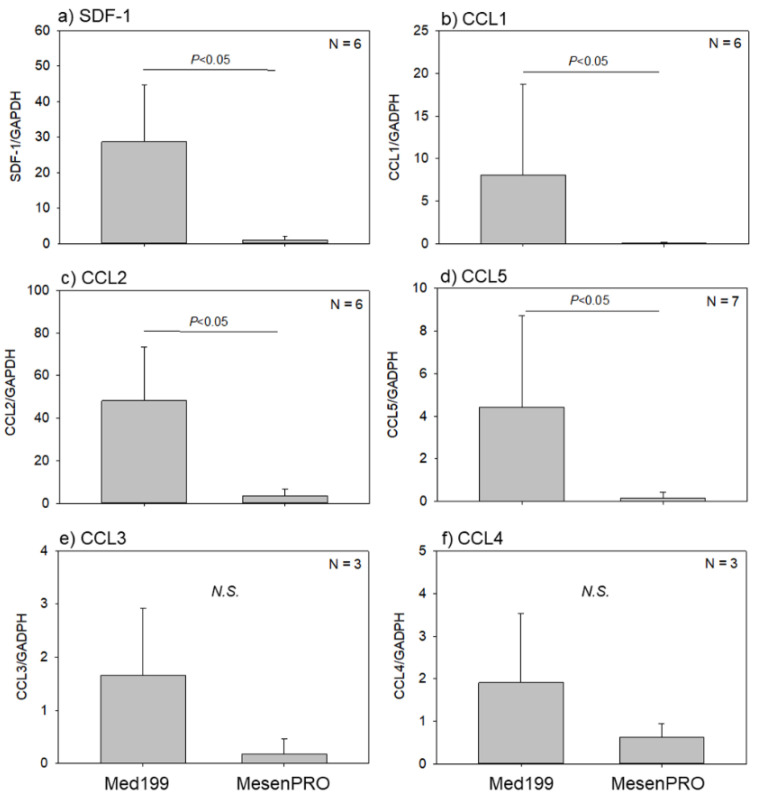
Effects of culture media on mRNA expression levels of chemotactic factors, (**a**) stromal cell derived factor-1 (SDF-1), (**b**) chemokine C-C motif ligand (CCL1), (**c**) CCL2, (**d**) CCL5, (**e**) CCL3, and (**f**) CCL4, in tissue-engineered periosteal sheets (TPSs) in vitro. *N.S.*: not significant.

**Figure 4 ijms-22-02169-f004:**
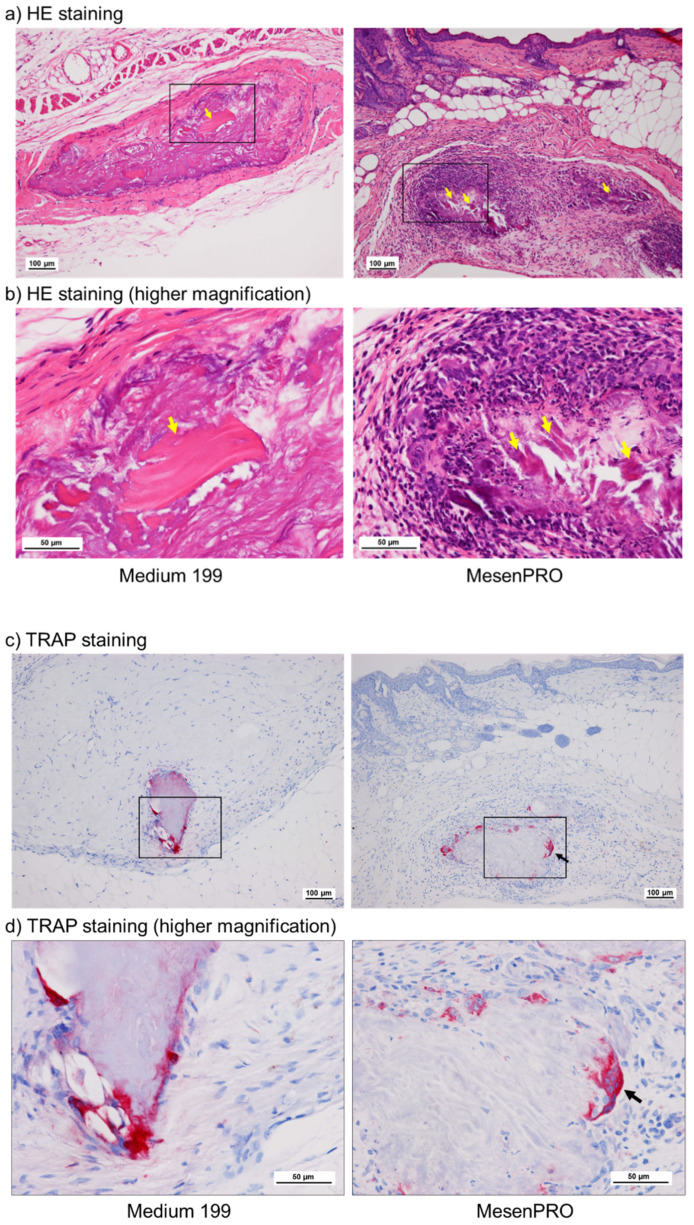
Effects of culture media on new bonelike tissue formation and tartrate-resistant acid phosphatase+ (TRAP+) cell induction in the implanted TPSs in vivo. (**a,b**) Hematoxylin and eosin (HE) staining and (**c,d**) TRAP staining. Panels (**b,d**) represent higher magnification of the corresponding regions shown in Panels (**a,c**). Yellow arrows represent bonelike tissues. TRAP+ cells were stained red. The black arrow indicates representative TRAP+ multinucleated cells (osteoclasts).

**Figure 5 ijms-22-02169-f005:**
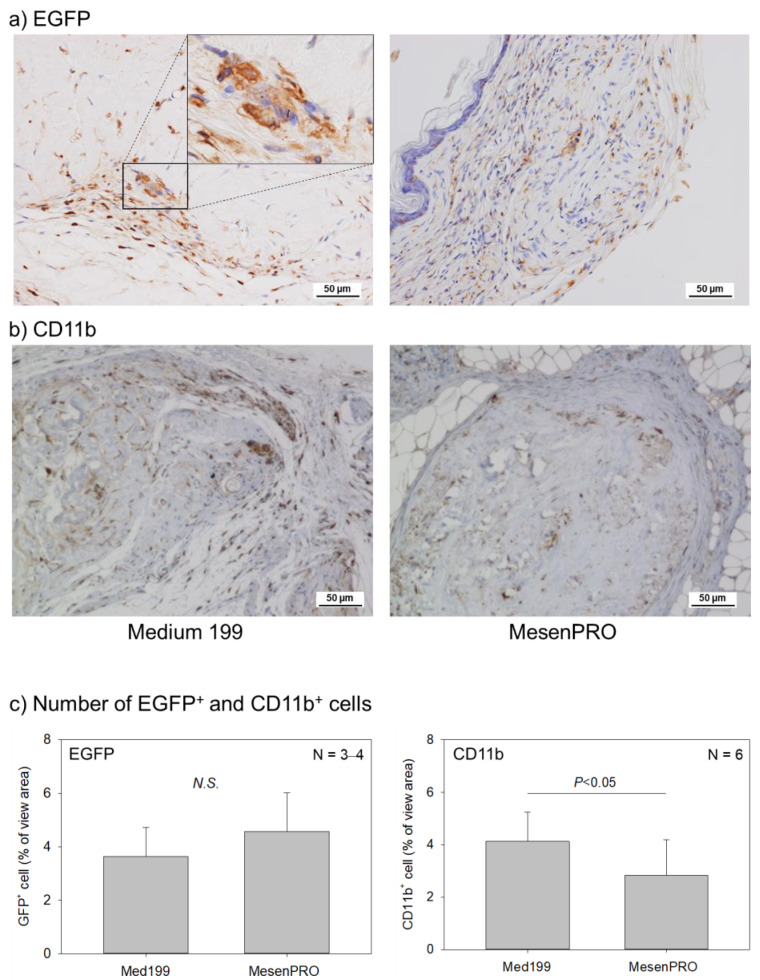
Effects of culture media on the infiltration of enhanced green fluorescent protein^+^ (EGFP^+^) and CD11b^+^ cells. (**a**) EGFP immunostaining (inset: higher magnification of the selected region), (**b**) CD11b immunostaining. Individual positive cells were stained dark brown. (**c**) Quantification of the immunohistochemical data. *N.S.*: not significant.

**Table 1 ijms-22-02169-t001:** Number of samples with new bonelike tissue formation.

	Total	Positive n (%)	*P* Value
Medium 199	13	6 (46.2%)	0.226
MesenPRO	12	9 (75.0%)	

**Table 2 ijms-22-02169-t002:** Number of samples with TRAP^+^ cell induction.

	Total	Positive n (%)	*P* Value
Medium 199	13	2 (15.4%)	0.097
MesenPRO	12	6 (50.0%)	

## Data Availability

The datasets used and/or analyzed during the current study are available from the corresponding author on reasonable request.

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
