# Peer review of "Osteoclastogenic Potential of Tissue-Engineered Periosteal Sheet: Effects of Culture Media on the Ability to Recruit Osteoclast Precursors"

_ijms, 2021, doi:10.3390/ijms22042169_

Round 1

Reviewer 1 Report

The manuscript entitled " Osteoclastogenic potential of tissue-engineered periosteal sheet: effects of culture media on the ability to recruit osteoclast precursors" is a significant study which provides useful information for future researches with impact on health fields. Current options in this field present limitations that lead to the need to expand and development of knowledge and, in this sense the data presented in this paper have impact on this problem, opening new directions. 

However, the manuscript needs some improvement in gaining the reader's attention. Here are some points that are needed to be addressed before publication.

  1. In the paper, the authors mentioned about the previous studies in order to compared data or to evidence the improving the study. From this reason, I think that the authors should better explain/ highlight the novelty of their work for a better understanding by readers. Certain ideas should be reformulated to clearly articulate their work.
  2. The authors should use/follow the journal guidelines, especially references, figures.

Author Response

1. In the paper, the authors mentioned about the previous studies in order to compared data or to evidence the improving the study. From this reason, I think that the authors should better explain/ highlight the novelty of their work for a better understanding by readers. Certain ideas should be reformulated to clearly articulate their work.

Response: Thank you for your comment. We have highlighted the novelty of this work by revising and restructuring the Introduction section.

2. The authors should use/follow the journal guidelines, especially references, figures.

Response: Nevertheless, we have revised the reference section. Also, the labels (a, b, c, …) have now been emboldened in the legends.

Reviewer 2 Report

Uematsu et al. described the different effect of traditional medium and stem-maintenance medium on human tissue-engineered periosteal sheets (TPSs). The aim was very interesting and experiment plan was well-designed. Anyway, some information in introduction and discussion are not relevant, and this may confuse the reader. Moreover, some minor and major inadequacies must be corrected:

  • Lines 57-63: it is not clear what the authors wanted to say. Firstly, they affirmed that TPS is composed by multi-layered periosteal cells in collagen matrix, but it does not form the well-reported structure. Furthermore, they talked about pluripotent stem cells and osteoblastic precursor cells, and then about periosteal stem cells again. The authors should be explained better the role of periosteal stem cells and the nature of TPS.

  • Line 70: it is not clear what is the stem-cell medium. Please, describe better the medium, indicating its role in stem maintenance and that it is not a stem-cell conditioned medium.

  • Lines 87-89: it is not clear the meaning of the sentence.

  • Magnification of figure 4a is not adequate to demonstrate the presence of bone-like tissue formation. Please, provide images at higher magnification that can demonstrate what the authors sustain. Furthermore, it is not identifiable TRAP+ multinucleated cells in both images of figure 4c.

  • Figure 5 shows PCNA and alpha-SMA staining in tissues that do not show also formed bone-like tissue, therefore it is not considerable to support cell proliferation and neovascularization the implanted TPSs in result discussion.

  • Lines 193- 207: it is not clear how this part is relevant for the discussion.

  • Lines 256-259: how is it possible that collagen, that is a protein, provides BMPs, other proteins? It is not clear what the authors mean.

  • The authors declared that they took periosteal tissue from the periodontal tissue, but they are not the same.

Author Response

- Uematsu et al. described the different effect of traditional medium and stem-maintenance medium on human tissue-engineered periosteal sheets (TPSs). The aim was very interesting and experiment plan was well-designed. Anyway, some information in introduction and discussion are not relevant, and this may confuse the reader. Moreover, some minor and major inadequacies must be corrected:

Response: We have deleted some portions of the Introduction section to focus on the matter investigated. In the Discussion section, you may be referring to the “General outlook” part. However, this subsection is important as it gives the readers a better understanding of the TPS characteristics, a biomaterial for bone regeneration. It also gives a background to TPS therapy. Therefore, we hope to keep it in the Discussion section.

- Lines 57-63: it is not clear what the authors wanted to say. Firstly, they affirmed that TPS is composed by multi-layered periosteal cells in collagen matrix, but it does not form the well-reported structure. Furthermore, they talked about pluripotent stem cells and osteoblastic precursor cells, and then about periosteal stem cells again. The authors should be explained better the role of periosteal stem cells and the nature of TPS.

Response: To highlight the novelty of this work, this portion has been deleted.

- Line 70: it is not clear what is the stem-cell medium. Please, describe better the medium, indicating its role in stem maintenance and that it is not a stem-cell conditioned medium.

Response: We have added a definition and an explanation of this stem cell medium.

- Lines 87-89: it is not clear the meaning of the sentence.

Response: This sentence was also deleted to highlight our purpose.

- Magnification of figure 4a is not adequate to demonstrate the presence of bone-like tissue formation. Please, provide images at higher magnification that can demonstrate what the authors sustain. Furthermore, it is not identifiable TRAP+ multinucleated cells in both images of figure 4c.

Response: In Panel (b) of Figure 4 (HE staining), we have added higher-power images of the bone-like tissue shown in the boxes within Panel (a). In Panel (d) of Figure 4 (TRAP staining), we have replaced the original images of TRAP+ cells with higher-power images of the cells. A typical multinucleated TRAP+ cell, a relatively mature osteoclast, was identified in the MesenPRO group. In the Medium 199 group, a multinucleated TRAP+ cell was also identified, but this cell contained fewer nuclei. Thus, it is probably a relatively immature osteoclast.

- Figure 5 shows PCNA and alpha-SMA staining in tissues that do not show also formed bone-like tissue, therefore it is not considerable to support cell proliferation and neovascularization the implanted TPSs in result discussion.

Response: We have deleted Figure 5 from the revised manuscript because these data do not significantly influence the main findings.

- Lines 193- 207: it is not clear how this part is relevant for the discussion.

Response: We believe that sound and functional bone regeneration can be achieved by well-balanced bone formation and resorption, a basic homeostatic process in bones. However, most bone substitutes used for reconstitution of small bone defects cannot directly induce osteoclast-dependent bone resorption. We think that this function influences the quality of newly formed “bone” and that this is the major advantage of TPS therapy. Therefore, we would like to retain this paragraph to support the significance of the data.

 - Lines 256-259: how is it possible that collagen, that is a protein, provides BMPs, other proteins? It is not clear what the authors mean.

Response: We intended to mean that collagen could function as a carrier of BMPs. In the revised version, we have slightly modified and reinforced this portion by incorporating some references.

- The authors declared that they took periosteal tissue from the periodontal tissue, but they are not the same.

Response: To avoid readers’ misunderstanding, we have added information on donor numbers, their genders, and the age range.